# Developing Return Supply Chain: A Research on the Automotive Supply Chain

**Liane Pinho Santos [1,*] and João F. Proença [1,2]**

[1] Faculty of Economics, University of Porto, 4099-002 Porto, Portugal; jproenca@fep.up.pt
[2] Advance/CSG, ISEG, University of Lisbon, 1649-004 Lisbon, Portugal
[*] Correspondence: lianepinhosantos@hotmail.com

**Abstract:** The purpose of this study is to investigate and discuss the challenges namely, the barriers and solutions to developing return supply chain policies in automotive industry. This industry has been suffering governmental pressure to achieve sustainability in all industrial processes. The solution is to reorganize the supply chain and rethink the product from the beginning (closed-loop supply chain evolving to cradle-to-cradle supply chain); however, the literature about this process is scarce. This paper presents exploratory research into the automotive supply chain in order to study the processes developed to achieve more sustainable supply chains. A strategic business net is used as theoretical sample to collect data. The results show that climate change concerns and governmental guidelines lead to sustainable organizational culture. It was found that sustainability is integrated in all processes, which affects business interaction with partners. The business interaction is difficulted by total product recycling. To overcome this barrier, innovation strategies, strategic alliances and governmental politics are presented as enhancers to return supply chain policies development. Organizational and cultural elements were presented as difficulties, but could be easily mitigated with the enhancers, alongside commitment and proactivity of companies. The research shows that when the challenges are overcome, the companies can integrate cradle-to-cradle design frameworks into their supply chains. This reorganization is necessary to achieve sustainability and accomplish governmental guidelines.

**Keywords:** sustainability; closed-loop supply chain; cradle-to-cradle supply chain; automotive supply chain

## 1. Introduction

Sustainability is not a new topic. For over 40 years, researchers have been warning about the impact individual and collective attitudes have on the environment [1].

During the year 2020 and the first half of 2021, with the world suffering the impact of the COVID-19 pandemic, sustainability issues became more evident. Through the confinement, much the world's population stayed at home, global industrial production decreased and some companies stopped working. With this impact, there was an air quality improvement [2,3]. The pandemic reduced the emission of greenhouse gases with the main relief being for the United States of America, China, and India, which together represent 60% of carbon emissions and affect climate globally [3]. During the confinement, there was a decrease in activity in these three countries, and in global totals. There was a reduction in air and road traffic, as well as a reduction in use of fossil fuels, which resulted in a sharp decrease in greenhouse gases for the environment. There was a 27.9% drop in $CO_2$ in April 2020 compared to the previous year. After that, with the so called "new normal", and economic return, industrial production restarted, with corresponding effects on climate change.

In August 2021, the Global Citizen Organization synthetized the main events related to climate change in the same year: in June, Canada and the Pacific Northwest of the United States of America experienced extreme heat, causing huge wildfires. Flooding occurred in Germany, Belgium, and China in mid-July. Heavy rains caused rivers to overflow in Uganda and landslides in India [4]. The planet is already 1.2 °C above the desired level and for that reason, the atmosphere gets warmer and retains more moisture, which causes more rain [5].

The pandemic helped people to realize the impact of the attitudes and behaviors of individuals and companies. With reduced consumption and air/road traffic, the "Earth Overshoot Day" was postponed 27 days in 2020 [6]. In this way, it has become clear that individual and collective changes can have positive effects on global sustainability [7].

It is known that the biggest impact on sustainability is caused by polluting industries, such as textiles, traffic, electrical, oil, and others [8]. These industries have been forced to be more sustainable and have practices better aligned with environmental concerns. The automotive industry will even have to be more sustainable, given the global policies such as the Paris Agreement, and the goal of carbon neutrality of the European Union [2,9]. The automotive supply chain includes multiple companies of different sectors, such as the vehicle producers and producers of components such as plastic, rubber, iron, glass, textiles and electronics [10,11]. European automotive companies have a guideline for recycle, reuse and recovery of their materials, namely, the ELV Directive (Directive on end-of-life vehicles—Directive 2000/53/EC—entry into force in 18 September 2000).

For these reasons, the automotive supply chain is changing the resources and materials used to accomplish the governmental guidelines and achieve sustainability goals. These changes are influencing the organization of the automotive supply chain. Innovation in resources, materials and processes are the key solution for accomplishing the governmental and other regulatory guidelines [11,12]. With the new policies and the necessity of rethinking the supply chain, multiple types of supply chain are presented as solution to sustainability issues, see Table 1.

**Table 1.** Definition of different types of supply chain presented as solution to sustainability issues.

| Types of Supply Chain Presented as Solution to Sustainability Issues | Definition | Authors |
|---|---|---|
| Circular Supply Chain | When a company focus on reducing their impact on the environment related to their products and rethinking the processes and activities considering the product's life cycle. | [13] |
| Reverse Supply Chain | The knowledge acquired in logistics—create structures to circulate the products and resources, develop a reverse supply chain within the closed-loop supply chain. | [14–20] |
| Closed-Loop Supply Chain | This supply chain considers all sectors and partners, creates, and organizes the network to recover, recycle and reuse its products. | [14–20] |
| Cradle-To-Cradle Supply Chain | The term cradle-to-cradle is a trademark of consultants William McDonough and Michael Braungart. This supply chain is based on the closed-loop supply chain with reorganization of the entire supply chain. The products are designed and manufactured with sustainable materials and safely returned, considering the technological and biological cycles in closed loops. Cradle-to-cradle is related to the total recycling through designing from the product start to finish. | [21–23] |

It is known that the automotive industry is applying the closed-loop supply chain to make all processes more sustainable and accomplish the Directive ELV [24]. Knowing this, for implementation of cradle-to-cradle supply chain it is necessary reorganization of supply chain and rethink the product from the beginning [25,26]. Developing the cradle-to-

cradle supply chain is a challenge, due to the need to reorganize the supply chain through innovation, and studies are still needed to facilitate this organization [12,27].

In this article, we will adopt the terminology "return supply chain" used in the Special Issue "Four Decades of Cradle-to-Cradle: The Contribution to Sustainable Supply Chains". This terminology refers to a closed-loop supply chain evolving to cradle-to-cradle supply chain.

With this challenge of developing a cradle-to-cradle supply chain and knowing the scarce literature about the theme in the automotive industry, we developed this research which aims to increase understanding of the issues, namely, the barriers and solutions to developing return supply chain policies in the automotive industry.

The organization of the paper is as follows: Section 2 presents the literature review and theoretical framework about climate change and sustainability in the automotive supply chain, and a bibliometric analysis on the different designations related to return supply chains. An exploratory study on a strategic business net of automotive supply chain is presented in Section 3, while the results are analyzed in Section 4. Section 5 presents the discussion. And finally, Section 6 presents the conclusions, limitations, and future research.

## 2. Literature Review and Theoretical Background

### 2.1. Climate Changes and Sustainability

Today, the population has experienced several divergent environmental phenomena associated with climate change. The constant evidence of global climate changes (natural disasters, forest fires, rising water levels, among others), shows that efforts to date have not been enough to promote sustainable development of the planet. Climate changes are defined as any change in weather patterns, mainly associated with greenhouse gases [7]—the main contributors to global warming. The increase in temperature of the planet is due to the excessive absorption of solar radiation and the reduction of heat radiation into space [3,28].

In 1997, the Kyoto Protocol was signed and was the first international agreement on controlling the emission of greenhouse gases. The protocol's objective was centered on a 5.2% reduction in the emission of greenhouse gases between 2008 and 2012, in parallel with a reduction in the environmental impact of companies' activities through sustainable development.

In 2015, the Paris Agreement was signed with the goal of limiting the global temperature increase to less than 2°C, through compliance with measures that limit or reduce greenhouse gas emissions by 55% by 2030 [29]. In parallel with the Paris Agreement, in the countries belonging to the European Union, the European Parliament has presented the European Green Deal, in which it aims to achieve carbon neutrality by 2050 [30]. The European Union believes that sustainability can be achieved through renewable energy, i.e., reducing dependence on fossil fuels (the main drivers of global warming), thus reducing climate change [31].

Sustainability encompasses social, economic, and environmental benefits, for individuals as well as for companies [32]. Sustainable development is the strategy used to achieve sustainability, that is, the company's entire strategy should consider current needs, without compromising future needs. Sustainable development focuses on the three areas of sustainability, namely, the environment, economic and social issues [30,33–35].

In line with governmental guidelines, also in the year 2015, the 17 Sustainable Development Goals of the United Nations were signed. The Sustainable Development Goals were agreed upon internationally with the goal of eradicating poverty, protecting the planet, and maintaining prosperity for all. The United Nations' goals promote sustainable development in the three dimensions of sustainability [36]. Through these goals, it is intended to integrate and apply sustainability in all companies, promoting sustainable development in the three dimensions of sustainability for society [36,37]. Specifically for

companies, they state that industries should aim to modernize their infrastructure and become increasingly sustainable using clean and environmentally friendly technologies and industrial processes. By accepting, respecting, and putting this commitment into practice, countries will achieve the balance between the three dimensions of sustainability.

*2.2. Sustainability in the Automotive Supply Chain*

Transportation is one of the sectors that contributes the most to the increase in greenhouse gases (air, road). Simultaneously with the increase in world population [38], it was found in 2018 that road transport was responsible for 13% of direct greenhouse gas emissions [2]. For this reason, the automotive industrial network is one of the main industries to suffer from stricter measures to reduce greenhouse gas emissions, since failure to meet the targets means heavy fines [38]. European Union directives are based on a set of standards to achieve carbon neutrality by 2050 [30,34]. These directives intend to substitute vehicles with lower emissions, and the electric vehicle is one of the current bets [39–41]. With this governmental influence, the automobile industry is being forced to change its resources and activities [40].

Portugal was the first European country to declare that it would be carbon neutral by 2050 [42]. According to data from the Ministry of the Environment and Climate Action, in 2019, Portugal had reduced emissions by 26%, despite economic growth above the eurozone. In that same year, Europe reduced emissions by 4.3% and Portugal reduced them by 8.5%, being the fifth-ranked country in respect of this goal: Estonia (−22,1%), Denmark (−9.0%), Greece, Slovakia (−8.9% each), Portugal (−8.7%) and Spain (−7.2%). The European Union considers Portugal one of the countries with the capacity to reduce its emissions, being aware of the associated climate challenges. Portugal is characterized as a "hotspot" for climate change [43,44]. Portugal accepts the European and international indications and the need for their local application. International cooperation is the answer to stop the increase in global average temperature and thus avoid the increase in climate change that we have been feeling. Portugal has several companies constituting the supply chain, including vehicle producers, suppliers, logistics and raw material [45].

In accordance with the changing context, the automotive supply chain manufacturers must define their strategy with sustainability in mind, adapting and focusing their processes based on sustainability. These changes in resources and activities will also affect the relationship between actors, that is, sustainability has influence on business interactions [11]. Interaction between businesses is not a short-term process, but leads to continuous relationships [46], creating interdependencies between companies [47]. The specificities of actors and their activities/ resources are an outcome and incentive for interaction [47]. Business interaction is "both a dynamic and a stabilizing force" [42] (p. 7), for creating a stable position in the ever-changing business world. The companies must interact and adapt their resources and activities to achieve sustainability. These actions are facilitated with open communication, trust, and commitment to positive interactions [11]. Such adaptation is easier in stable and longer-term business relationships [48–50].

Sustainable supply chain management characterizes supply chain management considering social, economic, and environmental issues [13,27,51–55]. Environmental sustainability is the sustainability dimension on which the European Union has focussed more attention. For that reason, Europe has undergone the most changes, with changes in resources that are able to reduce the impact on the environment. Sustainable supply chain management is about finding more sustainable solutions to reduce the impact on the environment while ensuring the bottom line, creating value for stakeholders, and meeting government targets [27].

Sustainability, embedded in an organizational culture, influences actors' beliefs and values related to guidelines, policies, and procedures [56]. Promoting a sustainability culture is a continuous process over time. When companies share their beliefs related their economic, social, and environmental aspects, it is possible to have aware and coherent organization action [57]. The automotive industry has been experiencing multiple

influences requiring change. A sustainability-oriented culture helps managers to accept the influence, create changes, embrace innovation, and maintain their changes, in the organization, in a sustainable way [56,58]. This occurrence is related to the bonds between behavior and culture. When there is reciprocity between culture and behavior, each one can be reinforced and shaped [57], which represents a behavior change accepted by a sustainable culture.

Supply chains are adapting their entire activity to reduce carbon emissions and reduce energy consumption and/or create renewable energy sources. The closed-loop supply chain concept focuses on material recycling since the product must create value even after use. Such reuse contributes to carbon emission reduction, simultaneously with increasing profit [19]. In the automotive industry there have been two ways of reusing products. There are companies that manufacture the products, but their partners reuse them—open-loop logistic. Likewise, there are companies that develop the product and are responsible for its end-of-life, i.e., collect and reuse the product in whole or in part—closed-loop logistics [14,17,18,27,59]. In the latter, the importance of total recycling of all products comes into play. In 2016, the materials of 91% of vehicles at their end-of-life, after being dismantled, were reused and recycled [60]. The need to recycle is due to the concern for sustainability, and to comply the European directives—2000/53/EC—whose objective is to reduce waste at the end-of-life of automobiles. Since 2015, there must be a minimum of 85% of reused and recycled materials in vehicles, and the automobile recovery rate must be 95% [61]. These goals require a supply chain reorganization and rethinking of the product design. Reorganizing the supply chain is a challenge due to the need to readapt activities and processes, establish new business relationships (if necessary), and address cultural aspects, such as resistance to change which delays restructurating, and lack of technological or knowledge support, as well as promoting a the commitment and resilience to achieve and put in practice the necessary changes [62]. Some authors point out innovation as the solution to reuse of all materials to ensure sustainability and adapt supply chain activities [12,63].

One of the solutions presented is to incorporate the cradle-to-cradle design framework into the supply chain. The cradle-to-cradle supply chain can go beyond total recycling, as it considers the technical and biological aspects of the products. Cradle-to-cradle approaches are related to total recycling through designing. The product is developed with a goal, namely, that the recycling costs are less than purchase of new material. For this reason, cradle-to-cradle methods become a competitive tool because they reduce costs [22]. In other words, cradle-to-cradle is a product design concept, with a certification system, focused on circularity and material health [22,63]. The main goal of the cradle-to-cradle supply chain is that industrial design must differentially process biological nutrients—biodegradable materials that should safely return to the environment—and technical nutrients—resources that are not continuously produced by the biosphere, such as metals and plastics, that would be continuously used in industrial processes without loss of quality. When applying cradle-to-cradle principles to industry, instead of thinking in terms of waste management or waste reduction, the very idea of waste is eliminated. The creation of products, with reorganization of design, proposes a future with resources availability [22].

Since 2005, companies can implement the "The Cradle-to-Cradle Certified™ Products Program" to help in the creation of products whose design ensures the health and circularity of products [64]. This program certificates the companies after verifying the development, the manufacturing, and the reuse potential of their products. This certification focuses on five categories: (i) Health Impact of Materials: ensures that the raw materials used in the manufacture of products are safe for health and the environment, using lower impact chemicals; (ii) Reuse of Materials: ensures that products remain in closed cycles through the reuse of materials, thus eliminating waste; (iii) Renewable Energy: ensures that renewable energy is used in the production process, which reduces/eliminates the impact of greenhouse gases on climate change; (iv) Water Management: identifies

water as a valuable resource, protects watersheds, and allows it to be available to people and all other organisms; and (v) Social Justice: ensures a dimension of respect for the labor and human rights of all those involved in the industry. Portugal only had one firm with this certification at the time of writing, which is not related to the automotive supply chain [65]. Cradle-to-cradle principles constitute a path toward achieving the United Nations' 17 goals [21].

Companies in the automotive supply chain, even if they do not seek certification, have invested in the near-total recycling of their components. The difficulties in recycling all the components are due to technical issues. For this reason, developing the cradle-to-cradle supply chain is a challenge, due to the need to reorganize the supply chain through innovation, and studies are still needed to facilitate this reorganization [12,27].

### 2.3. A Bibliometric Analysis about Supply Chain Related Concepts for Sustainability

The reorganization of supply chains is necessary to accomplish the governmental guidelines, as well to decrease industries' impact on the environment.

The literature shows multiple types of supply chain as solutions to sustainability issues.We verify several authors giving different designations for concepts, such as "circular supply chain", "closed-loop supply chain", "reverse supply chain" and "cradle-to-cradle supply chain". All these concepts are similar and show diverse solutions to implement sustainability in the supply chain, as shown in Table 2.

**Table 2.** Different types of supply chain related to sustainability issues.

| Concept | Definition |
|---|---|
| Circular Supply Chain | "Coordinated forward and reverse supply chains via purposeful business ecosystem integration for value creation from products/services, by-products and useful waste flows through prolonged life cycles that improve the economic, social and environmental sustainability of organisations." [47] (p. 447). |
| Closed-Loop Supply Chain | "Closed-loop supply chains integrate reverse logistics and forward logistics by collecting and remanufacturing used products to reduce production costs and exercise environmental responsibility" [16] (p. 2). |
| Reverse Supply Chain | Closed-loop supply chain involves the "reverse supply chain" [17] which links to reverse logistics and "implies coordination with customers (of the forward supply chain) and subsequent activities beyond the transportation and storage of materials such as material recovery or recycling" [13] (p. 199). |
| Cradle-To-Cradle Supply Chain | This supply chain is based on product design concepts created to eliminate waste, where products are designed and manufactured from sustainable materials and safely returned to the technical and biological cycles in closed loops [22]. |

We have done a bibliometric analysis about subjects related to sustainable supply chains in the automotive industry. The bibliometric analysis was made in the Web of Science (WoS) database in August 2021. We chose the WoS because it is a comprehensive and multidisciplinary research platform which allows researchers to search various terms and data. The WoS database is widely used in bibliometric analysis and has a very good reputation in the scientific field [57]. To search we used keywords related to the different designations about the issue adding the keywords "sustainability" and "automotive". The time range selected was from the beginning of 1970 to the end of August 2021 because the term "cradle-to-cradle" in this sense was introduced around 1970 by Walter Stahel [58]. The results of the bibliometric analysis are shown in Table 3.

**Table 3.** Summary of bibliometric analysis.

|  | Keyword(s) Used | Number of Articles | Total Number of Articles | Number of Repeated Articles [1] |
|---|---|---|---|---|
| Articles related to different designations for supply chain | "Reverse supply chain" | 545 | 2458 | 128 |
|  | "Circular supply chain" | 77 |  |  |
|  | "Closed-loop supply chain" | 1538 |  |  |
|  | "Cradle-to-cradle" | 298 |  |  |
| Articles related to different designations for supply chain + sustainability | "Reverse supply chain" and "sustainability" | 103 | 566 | 23 |
|  | "Circular supply chain" and "sustainability" | 44 |  |  |
|  | "Closed-loop supply chain" and "sustainability" | 280 |  |  |
|  | "Cradle-to-cradle" and "sustainability" | 139 |  |  |
| Articles related to different designations for supply chain + automotive | "Reverse supply chain", "automotive" | 8 | 36 | 1 [21] |
|  | "Circular supply chain" and "automotive" | 0 |  |  |
|  | "Closed-loop supply chain" and "automotive" | 25 |  |  |
|  | "Cradle-to-cradle" and "automotive" | 3 |  |  |
| Articles related to different designations for supply chain + sustainability + automotive | "Reverse supply chain", "sustainability" and "automotive" | 5 | 14 | 0 |
|  | "Circular supply chain", "automotive" and "sustainability" | 0 |  |  |
|  | "Closed-loop supply chain", "automotive" and "sustainability" | 8 |  |  |
|  | "Cradle-to-cradle", "automotive" and "sustainability" | 1 |  |  |

[1] This column demonstrates the number of repeated articles in 4 keywords researched.

First, it is possible to observe significant research related to multiple types of supply chain. We found in total 2330 papers (e.g., 2458-128), which decreases to 543 (566-23) articles if we focus on sustainable supply chain (less 1787) and decreases even more if we focus on automotive: 35 (36-1, e.g., less 2295). When we searched the keywords all together, we got a total of 14 articles. With this, it is possible to show that despite different designations for supply chain, there are only a few articles published about the sustainability impact in any type of automotive supply chain. The 14 articles discussing sustainability in automotive supply chain are presented on Table 4. These articles were published between 2010 and 2020 in diverse journals, although four of them were published in the Journal of Cleaner Production.

After full reading of the 14 articles, it is possible to observe that only four define any type of supply chain [14–16,20]. Some of them present literature reviews related to sustainability in the automotive supply chain and suggest further research on the field. The articles show different approaches for the return supply chain and mostly refer to barriers for return supply chain implementation and not solutions. These papers divide barriers as internal and external. As external barriers, Lemtaoui & Oueldrhiri (2019) referred to appropriate environmental regulations, uncertainty about the results, and lack of awareness of the partners about the return supply chain [14]; Gunther et al. (2015) noted governmental aspects and the transformation process to electric cars [66]; El Baz et al. (2018) cited a lack of tax incentives, together with costs, and unpreparedness of suppliers [16]; and Erol et al. (2010) referred to legislative issues [67]. As internal barriers, the same authors refers to lack of training and qualification of employees, lack of management commitment, and lack of information systems and technology [14]; lack of experience of the employees, lack of skills and difficulties to relate to their partners [16]; and lack of an information system, infrastructure, and technology [67]. Moreover, our bibliometric analysis evidences the literature gap and the need to research the reorganization of sustainable supply chains in the automotive industry.

As previously mentioned, the return supply chain is a current necessity and obligation due to climate issues and governmental influences, but companies have not yet managed to fully apply it. This may be related to the lack of studies [12,27]. Some authors suggest that the redesign of the supply chain to achieve sustainability is possible through innovation. However, we still do not understand the challenges faced in developing return supply chain policies in the automotive industry.

**Table 4.** Articles related to sustainability in the automotive supply chain.

| Authors | Title of the Article | Source |
|---|---|---|
| [16] | Reverse supply chain practices in developing countries: the case of Morocco | Journal of Manufacturing Technology Management |
| [67] | Exploring reverse supply chain management practices in Turkey | Supply Chain Management-an International Journal |
| [66] | The role of electric vehicles for supply chain sustainability in the automotive industry | Journal of Cleaner Production |
| [68] | Life cycle impacts of three-way ceramic honeycomb catalytic converter in terms of disability adjusted life years | Journal of Cleaner Production |
| [15] | In support of open-loop supply chains: expanding the scope of environmental sustainability in reverse supply chains | Journal of Cleaner Production |
| [14] | Reverse supply chain practices in the Moroccan automotive industry: an exploratory study | International Colloquium on Logistics and Supply Chain Management (LOGISTIQUA) |
| [69] | A multi-objective, multi-product and multi-transportation mode sustainable closed-loop supply chain network design | International Conference on Logistics, Informatics and Service Sciences (LISS) |
| [70] | An expert fuzzy rule-based system for closed-loop supply chain performance assessment in the automotive industry | Expert Systems with Applications |
| [71] | Multi-objective optimization of closed-loop supply chains in uncertain environment | Journal of Cleaner Production |
| [72] | Enhancing purchase intention in circular economy: empirical evidence of remanufactured automotive products in Thailand | Resources Conservation and Recycling |
| [73] | Environmental and economic assessment of closed-loop supply chains with remanufacturing and returnable transport items | Computers & Industrial Engineering |
| [74] | Vision-based identification service for remanufacturing sorting | Sustainable Manufacturing |
| [75] | Factors influencing the purchase intention of consumers towards remanufactured products: a systematic review and meta-analysis | International Journal of Production Research |
| [20] | Closed-loop supply chain planning model of rare mmetals | Sustainability |

## 3. Research Methodology

The purpose of this study is to investigate and discuss the barriers and solutions to developing return supply chain policies in the automotive industry.

For primary collected data, we read public and available secondary data related to businesses that belong to the automotive supply chain, such as reports, investigation papers and other documents meaningful for this investigation. With this, we could understand the context of the businesses in the automotive supply chain and gain knowledge necessary for the interviews.

Moreover, to address the research questions, we have designed an exploratory study to understand the phenomena under research using a strategic business net as data theoretical sample.

### 3.1. Research Design and Data Collection

In our desk investigation, we observed that in Portugal there exist four Original Equipment Manufacturers (OEMs): the final vehicle producer and several suppliers' companies responsible for components such as rear-view mirrors, navigation panels, plastic, paints, textiles, rubber, tires, lighting systems, belts, among many others. Being aware of the variety of companies involved, we decided to collect this data for the Portuguese automotive industry supply chain, which was considered to be a very rich and adequate option for our exploratory research. Then, we contacted multiple companies of the different sectors operating in the automotive components supply chain.

The selection of companies for data collection was based on the following criteria:

(i)     the company should be based in Portugal and/or should have operations embedded in the automotive Portuguese supply chain;

(ii)    the company must have a sustainability department;

(iii)   the company must have public sustainability reports;

(iv)    the company must have a Quality Management System certificate (ISO 9001) or an Environmental Management System certificate (ISO 14001).

These criteria ensure the inclusion of companies with certified sustainable practices, excluding companies that do not base their activity on sustainability. In addition, public access to sustainability reports helps researchers to learn about the company, which is necessary for empirical data collection and interviews. After the contacts, and being aware of the pandemic situation, we asked our interviewees to refer us to other contacts. With this recommendation and taking into account the four previously noted criteria, we chosepurposeful sampling [76] as a qualitative investigation technique. This technique is used in "identification and selection of information-rich cases for the most effective use of limited resources" [68] (p. 2). With this technique, we were able to select businesses with relevant knowledge and experience. The type of purposeful sampling used was snowball, which allows us to identify other businesses of interest with knowledge and/or similar characteristics that are valuable for this investigation [76].

The identification of other companies that have knowledge of the investigation issue, leads us to use a strategic business net [77] as theoretical sample, which is constituted by twelve business firms. These twelve companies selected are suppliers of the OEMs and are related to each other. These companies form a strategic business net, which means a group of actors sharing the same goals. According to the literature, the goals of the strategic business nets are more important than the individual goals [77–79]. The major goal of this strategic business net, shared by these twelve companies, is to increase their presence in the automotive industrial network globally.

A strategic business net is defined as a cooperative network created based on shared goals, in which the actors share their knowledge, resources and financial support to achieve competitive positions. The global goals are important, and the activities and functions are defined previously. The strategic net only exists when the actors recognize the global goals and are prepared to cooperate to achieve them. The strategic business relationship is characterized by cooperation, stability, and change. This characterization leads to dynamic evolution [80].

Businesses intentionally establish a network strategically in order to make business together and establish specific activities undertaken by the actors [80]. The actors have an important role, producing, delivering and organizing activities and resources [78]. The companies maintain stable boundaries between companies, focusing on functions and responsibilities. The density, multiplicity, reciprocity of ties and shared values distinguish this type of business network [77]. The strategic business nets are established through business relationships, and they are manageable [80]. With this, changes in network can be seen and diffused by other partners [80]. For this reason, business interaction is influenced by organizational change and influences the dynamic of strategic business net. This dynamic consists in a complex pattern of activities and business interaction. Through

these activities and interactions between the actors in the network, the dynamics emerge, where we can observe changes in the characteristics of the network, that is, in the structures, relationships, actors and their roles [80].

The companies researched are represented with numbers on Figure 1, where the OEMs are the final vehicle producers. The firms represented with a number inside a circle are suppliers of modules and systems; the companies represented through triangles are suppliers of components; and the companies represented with squares are suppliers of raw materials and other standardized parts. Table 5 presents the twelve companies and the functions of the interviewees involved in our data collection.

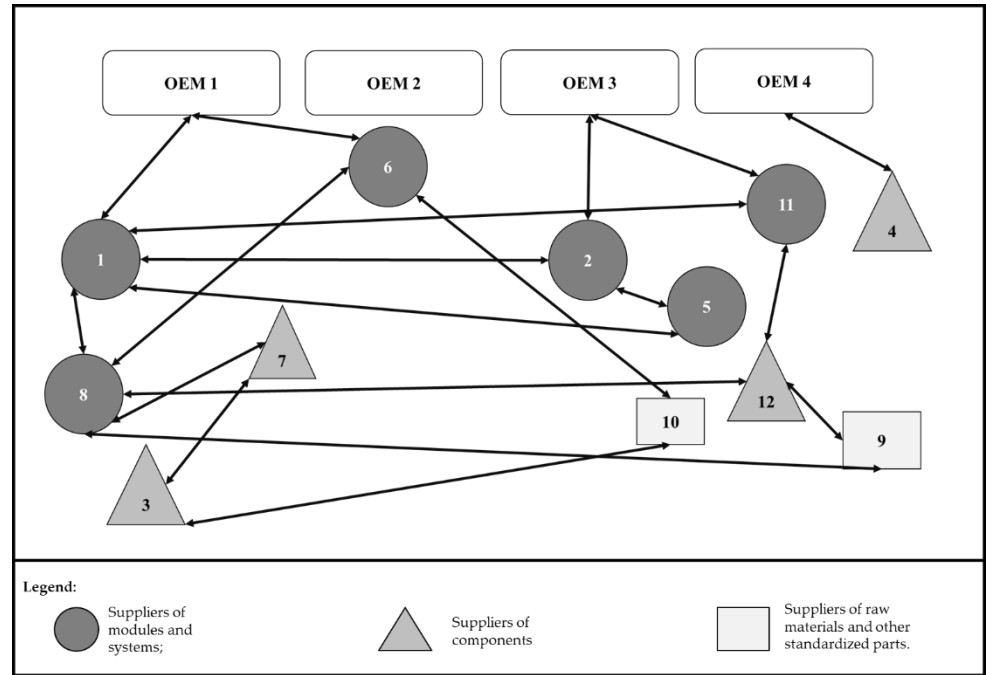

**Figure 1.** Strategic business net from data theoretical sampling.

**Table 5.** Identification of companies and interviewees.

| Companies/ Interviewee | Activities | Interviewee Position |
|---|---|---|
| 1 | Engineering services | General Director |
| 2 | Automation services | Innovation Engineer |
| 3 | Windshields and mirrors supplier | Managing Director |
| 4 | Emblems | Innovation Engineer |
| 5 | Multimedia supplier | Internal Control Coordinator Responsible |
| 6 | Tire supplier | Human Resources Director |
| 7 | Plastics systems support material firm | Marketing Manager |
| 8 | Interior and cockpit supplier; Sustainable mobility | Innovation Engineer |
| 9 | Foams and felt for sound insulation firm | Quality Engineer |
| 10 | Vehicle plastic and metal supplier | Purchasing Manager |
| 11 | Vehicle electronics supplier | Logistics Director |
| 12 | Technical parts for vehicle interiors firm | Human Resources Director |

The interviews to collect the data were conducted through two phases both via telephone calls and by the Zoom platform since the research took place during the period of the COVID-19 pandemic. The interviews were semi-structured and semi-directive, taking into account the script presented in Table 6. The global goal of interviews was to understand the barriers and solutions to developing return supply chain policies in the automotive industry. We started with questions about the meaning of sustainability for the company and the reasons for its implementation. After that, we asked how sustainability was verified in the three dimensions internally, in the company, and externally, with the

companies in the supply chain. Following this, we verified the solutions and barriers to being fully sustainable, that is, to implement the return supply chain. The collection of primary data was made in two phases. The first phase lasted from October 2020 to February 2021. Later, we developed a second phase to collect data focused on the cradle-to-cradle supply chain, using the same interviewees. The second phase took place between July of 2021 and September 2021. This phase was intended to confirm the previous results—the barriers and the solutions—and to ask additional questions needed for our analysis of the redesign of supply chains.

**Table 6.** Interview script.

| Interviews Script |
| --- |
| Phase 1 (October 2020–February 2021): |
| • What does sustainability mean for the company? What are the goals of sustainability for the company? |
| • Reasons for implementing sustainability in the company. (If referred to, explore public policies) |
| • Main concerns of the company's activity with the need for sustainability. |
| • How is sustainability being implemented? |
| • Implementation of sustainability—3 pillars? Describe. |
| • Effects spread by partners in the supply chain. |
| • Changes in activities by sustainability? |
| • Limitations to being "fully" sustainable. |
| • Associated difficulties in implementing sustainability. |
| • Possible solutions to implementing sustainability. (If mentioned, explore innovation) |
| • Other relevant matters. |
| Phase 2 (July 2021–September 2021): |
| • Has sustainability caused change? Describe. |
| • Have the difficulties presented been overcome? |
| • Have the solutions presented been implemented? Yes/No? explain |
| • Implementation of cradle-to-cradle in the activities. |
| • Difficulties in the implementation of cradle-to-cradle supply chain. |
| • Solutions for the implementation of cradle-to-cradle supply chain. |
| • Other relevant matters. |

*3.2. Data Analysis*

After data collection, we used Bardin to develop content analysis [81]. The content analysis was undertaken in three steps: analysis organization (pre-analysis and exploration of data), coding process (identification of keywords, themes) and categorization (establishment of criteria for categories and identification of categories). Figure 2 represents the sequential process of categories selection.

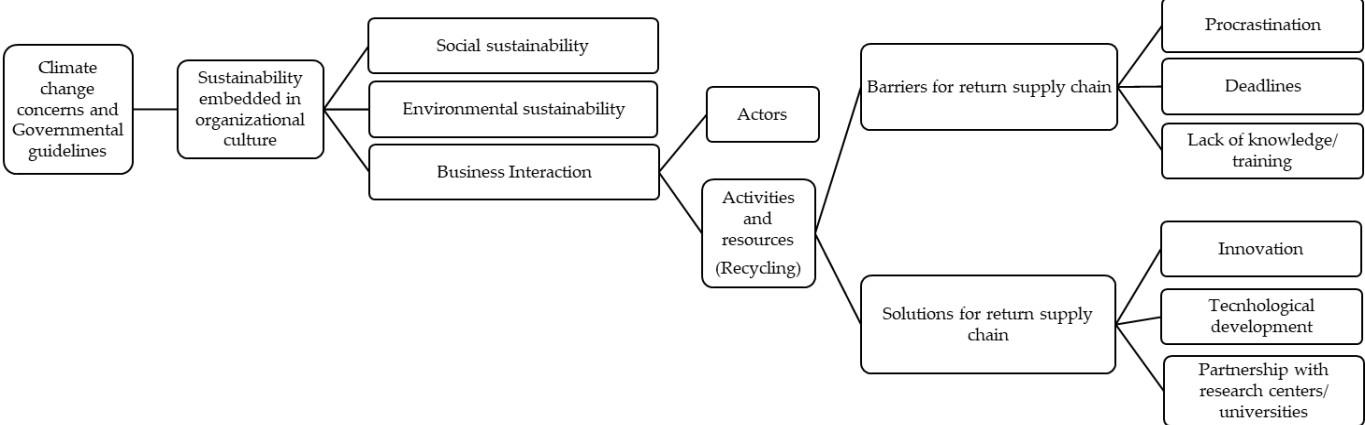

**Figure 2.** Main theme, categories, and sub-categories of data collection.

The coding process was based on meaning categories. Exploring the data allowed for the creation of a general idea of the data. Through this, the data was grouped by similar themes and meaning categories. This organization allowed a sequential process of data analysis, in terms of results and understanding of the data. The realization of a sequencing of the data allowed for a meaningful structure of the information.

In the first analysis, we tried to verify the external influences for the companies. In this regard, we verified the importance assigned to "Climate change and governmental guidelines". These external influences ended up shaping the companies, and we verified the category "Sustainability embedded in organizational culture". Through this category, we verified information about different forms of sustainability present in the companies, and the categories "Social sustainability", "Environmental sustainability" and "Business interaction" emerged. It is in the latter category that the data was concentrated. The interviewees mentioned several changes in business interactions both between categories "Actors" and "Activities and resources". However, it was in the second category that the main challenges for the implementation of the return supply chain emerge. The challenges were divided into two categories "Barriers to return supply chain" and "Solutions to return supply chain".

The results of the coding process reflect the authors' own interpretation of the information given. After our interpretation and separation into categories, it was presented to the interviewees for validation.

## 4. Results

After the content analysis of our data, it was possible to understand and discuss the challenges for automotive companies developing return supply chain strategies and policies. This section presents the main results of our research.

### 4.1. Climate Change Concerns and Governmental Guidelines

The interviewees stated that climate change and governmental directives are leading to changes in the companies. Furthermore, the influence of the global automotive industrial network in promoting sustainability shapes the Portuguese automotive supply chain performance to be in line with other partners. "*We are aware of everything that happens worldwide [regarding sustainability]. We can't forget especially what the European Union is indicating, as well as, on a global level like the Paris Agreement. With these kinds of guidelines, we really have to change our way of working.*" (Interviewee of company 10). "*Climate change, the influence of the global industrial network, the economic demands, the social demands and being aware of these global and environmental threats that we have been feeling, are big influences for our industry. Our awareness before was already on alert, now it's even worse.*" (Interviewee of company 2).

The challenges and barriers faced by Portuguese automotive industry are the same as the global automotive industry. Portugal has some specific guidelines—the European guidelines, referred to previously, since Portugal belongs to the European Union and has national legislation such as as "Lei de Clima". This legislation provides targets and instruments that lead to an end to the use of vehicles with combustion engines, in addition to other measures such as the establishment of zones within cities, including but not limited to historic centers, with severe restrictions on private car use.

The necessity of sustainability in present and future actions was emphasized, as well as its importance in the entire management and corporate culture. The interviews and all the information provided through sustainability reports and internal documents, support the concerns and relevance of sustainability for the companies. "*Sustainability, (…), became one of our pillars, our focus. It is a mission, not of the future, but of the present. We really must change the present, to present solutions now, for a better future. (…).*" (Interviewee of company 11).

Due to sustainability concerns and governmental directives, the companies are applying sustainability in all processes. The modification of the processes was guided by the 17 United Nations Sustainable Development Goals, highlighting specifically 9 of the 17

goals: "Clean water and sanitation", "Affordable and clean energy", "Industry, innovation and infrastructure", "Sustainable cities and communities", "Responsible consumption and production", "Climate action", "Life below water", "Life on land", and "Partnerships for the goals", which are more relevant to the automotive supply chain. "*One of our major goals is to continue to ensure that the group does act globally and that we are respecting the United Nations' sustainable development goals.*" (Interviewee of company 8). "*We've been continuing with [United Nations sustainable development goals] and understanding whether we've really been able to maintain them over time. The goal is to do a follow-up mainly at the social level.*" (Interviewee of company 11).

### 4.2. Sustainability Embedded in Organizational Culture

The development of sustainability policies was based on sustainability as a corporate culture. For this reason, sustainability effects were spread to social and environmental issues, as well as influencing the business interactions with other partners in supply chain.

Sustainability being the company culture, it was found to be integrated into all activities, shaping the company's mission and values. "*Sustainability for us is a commitment. We know that we are not an isolated reality and that we must be aware of our surroundings. (…)*" (Interviewee of company 2). "*(…) Sustainability has to be transversal to all our activity. Sustainability is integrated in our culture.*" (Interviewee of company 10).

It was possible to demonstrate that companies have been making their internal practices more sustainable as well. We noted a concern with the resources and material used internally, mainly at the administrative level, for example, in the reduction of paper, use of reusable bottles, heating of offices and sustainable mobility for employees.

#### Social Sustainability

Still on internal processes, we observed a concern with social sustainability (of employees and places where companies are sited). "*As much as sustainability is with a material focus, we are always involved in social transformation projects. We are completely against discrimination, unequal opportunities, and insecurity in the workplace. Above all when people share our values and goals, we all walk together (…)*" (Interviewee of company 2). "*(…) We plant the trees; we also clean the rivers. Thus, we reduce our environmental and local impact. This way, the population, as well as the companies, are benefited. (…) It was something very positive, since the community felt involved (…).*" (Interviewee of company 11).

#### Environmental Sustainability

Regarding environmental sustainability, the companies focus on reducing the impact of their activities on the environment where takes place. This reduces impact globally since it is also necessary to comply with governmental guidelines on such issues. Environmental sustainability is the sustainability dimension on which the European Union focuses more attention. For that reason, it has led to the most changes since the changes in resource used are able to reduce the impact of their activities on the environment.

To reduce the impact of their activity on the environment and achieve environmental sustainability, we observed several aspects: reduction in the consumption of fossil fuels, which reduced carbon emissions, a reduction in waste production (by recycling and reusing products), greater efficiency in water consumption (along with the preservation of biodiversity by reducing polluting chemicals), and energy efficiency. "*We have reduced fossil fuel consumption by 43% and increased electricity consumption by 20%. We achieved greater efficiency in water consumption, carbon dioxide emissions, waste produced (…)*" (Interviewee of company 2). "*(…) We have tried to manage everything that are chemicals that have a negative impact on the environment, (…) and we have used resources more efficiently and also water use more efficiently.*" (Interviewee of company 11).

In data collection we found that companies have been investing in the production of their own renewable energy, to ensure their energy efficiency—meeting one of the goals of sustainable development, namely, "Clean and affordable energy". Our results show that companies are producing their own energy, mostly solar energy, through

photovoltaic panels. In some situations, companies establish new partnerships to achieve this strategic goal. *"We are a large industry and for that reason we have started to take advantage of the size of the company. In this way, we have already been able to produce renewable energy on our premises through solar panels."* (Interviewee of company 5). *Some companies, while seeking to achieve energy efficiency and the reduction of carbon emissions in car production and car after-market, have done carbon offsets and environmental reforestation projects.* "*We are moving towards climate neutrality, as well as energy efficiency (…) In parallel, we make carbon offsets and are creating and/or developing new forms of energy (...).*" (Interviewee of company 5). "*(…) we launched a reforestation project because we started to see the consequences more and more tightly [of the activity] (…)*" (Interviewee of company 11).

### 4.3. Sustainability and Business Implications in Industrial Processes

The organization's culture conditions interaction with business partners. Sustainability and associated changes caused reorganization of the supply chain with respect to business relationships with partners (the actors), as well as with the resources used and activities developed. These aspects are the most influential for the application of a return supply chain. Next, we analyze in detail the implications for actors, activities, and resources.

#### Actors' interaction

The reorganization of supply chains modified the way actors interact. We found that the established relationships were based on a set of rules and/or requirements about sustainability. In the already established relationships these rules are not so evident, since the relationships that are maintained are due to common objectives and values. However, some interviewees mentioned that the prior definition of (public) rules for future suppliers helps to keep away and reduce the time lost with not so interested partners. With this, the relationships established have less risk of conflict, since they share and have accepted the same rules and requirements. "*(…) we defined a set of rules/requirements for our future suppliers. Initially, it was a little difficult, but then we ended up getting that whoever wanted to establish a relationship, was willing to comply with the rules (…).*" (Interviewee of company 3). "*The companies that are around us have to share [the same goals] and it is a must that our partners have an activity with social responsibility and environmental preservation.*" (Interviewee of company 2). Furthermore, the common goals among partners and the application of the rules for the new relationships demonstrate benefits, such as cost reduction, efficiency, quality assurance, increased competitiveness, and value creation for all. "*One of the big focuses of our work is competitiveness, that is, to be able to guarantee efficient processes along with a reduction in costs, in addition to benefiting our partners (…).*" (Interviewee of company 9). "*Certifications are also constant, that is, we maintain and guarantee quality management.*" (Interviewee of company 9). "*Sustainability means improving our work and facilitating the work of our partners. (…). We are focused on reducing time in the development of parts (…)*" (Interviewee of company 1).

As it was possible to demonstrate, sustainability influences the business interactions, whether internal or external processes. The dynamic of business interaction is powered by the sustainability culture in which it is inserted. These links show that sustainability goes across business boundaries. Ford et al. (2010) report that business interaction is a process that affects the direct relationship between companies, and indirectly affects all other business relationships, and has impacts more broadly across the network [47].

#### Logistics and reorganization of supply chain: activities and resources

One of the most repeated aspects in the interviews was the strong concern with logistics. Companies state that the greatest economic benefits—cost reduction—and environmental benefits—reduced carbon emissions—also pass through logistics. In Portugal, some companies work in business parks (as it happens near AutoEuropa) and this way, transport emissions are low, and not a major concern. During the relocation of companies or merger of companies, the solution is presented as reducing carbon emissions between partners.

*We have to make sure that our partners also stay the course. It's no good having a green product if my logistics are not green. So, currently, our biggest goal is a reorganization of the industry itself, as well as, to integrate ourselves into the industrial parks with the companies that produce vehicles. (…) It's something that will have to happen to the automotive industrial network, reorganize and join.* (Interviewee of company 5).

Sustainability also caused reorganization of the supply chain with respect to resources and activities. The changes in business interaction are focussed on vehicle production. To achieve the governmental guidelines, the automotive companies have changed their resource use and products. Most companies are focused on recycling and reusing products to execute a return supply chain. The life cycle assessment is a tool used by companies to understand all the risks associated with product life. "*We have been doing life cycle assessment, that is, an evaluation of our products since 2009. (…) With that, we were able to realize that many of the resources can be other resources and what we can transform them into. (…).*" (Interviewee of company 10). Most companies are only performing product recycling, without changing the product design, which will help ensure a closed loop without difficulty. However, some companies are showing concern that they will not be able to realize full product recycling, and that it is important to provide a solution to the present and existing resource use and thus waste. "*One of our goals is to get 100% of our materials recycled. By 2020 we have achieved 80%. We want to increase so that we can achieve a true circular economy, while being aware that it can take years*". (Interviewee of company 12). Respondents noted that some products lose quality after recycling and companies find it difficult to guarantee the same quality after reuse to their partners. Some interviewees mentioned that recycling may not be the definitive solution for the future. "*(…) in recycling, most products end up losing quality. In that way, recycling is not a long-term solution and recycling may only be a transitional solution and not be that definitive solution.*" (Interviewee of company 6).

### 4.4. Challenges for Return Supply Chains

As way to develop a return supply chain, companies have invested their attention to recycling. However, as mentioned before, they have had difficulties in product recycling due to the inability to recycle products in their totality, as well as quality assurance. In addition, they mention barriers for return supply chain implementation such as procrastination, deadlines, and lack of knowledge and training. On the other hand, the companies also presented some solutions to overcoming the barriers, specifically, innovation, technological development, and research partnerships.

The interviewers cited some barriers that hinder the development of return supply chain:

- Procrastination;
  - In some interviews, they referred this way of acting as "natural in Portuguese culture", that is, the people are resistant to change;
  "*Change is also difficult. We are integrated in a society that finds it difficult to change. (…).*" (Interviewee of company 7).

- Time issues related to deadlines for sustainability implementation;
  - "*The only problem with change is the time issue. When we are asked for "changes for yesterday" and we don't have support it is very difficult.*" (Interviewee of company 10);

- Lack of knowledge and training to reorganize the supply chain;
  - "*Knowing the influence of the European Union, being aware of the measures and the need to reduce the climate impact, we need to implement measures related to climate change. However, we still need help in terms of training to be able to implement and plan national strategies to reduce our influence on climate change. (…)*" (Interviewee of company 2).

These challenges were mentioned as the main barriers to the implementation of a return supply chain. Each challenge individually proves to be a limiting factor. However, in some companies we observed all three challenges together, which proved to be an increased challenge. For this reason, the companies, despite accepting the need to be sustainable, still do not make all their processes sustainable. They keep their focus on the closed-loop supply chain, that is, on recycling their products. Their lack of knowledge makes it impossible for companies to modify products from the start, i.e., what is needed for cradle-to-cradle supply chain implementation.

On the other hand, the interviewees mentioned ways to deal with the challenges. They cited the following solutions to overcome the previous barriers for implementation of the return supply chain:

- Innovation of processes and products;
  - ○ *"Innovation is the key to be able to achieve this sustainability in the whole process."* (Interviewee of company 2) *and "Our group is based on the innovation strategy. Innovation is trying, researching, trying, developing, trying, researching, and, only then, through these many attempts at error, the group can develop and experiment with new processes, new activities, new equipment. We have test centers and development centers in the laboratory that are helping us to reach the objectives to modify [the products] (…)"* (Interviewee of company 4);

- Technological development;
  - ○ *"Technology has definitely helped us a lot to achieve innovative solutions in all processes or in all aspects that encompass our work."* (Interviewee of company 9).

- Necessity of research and establish partnership with research centers and universities;
  - ○ *"We have been making an investment in research and asking for help from some research centers and our partners that will help us to be even more sustainable (…)."* (Interviewee of company 9) *and "The involvement with universities and research centers has helped us a lot to be able to maintain our position in the value chain."* (Interviewee of company 10)

These solutions are interconnected. Companies with high financial turnover have the internal capacity to conduct research and/or acquire knowledge through partnerships with other companies. This initial investment facilitates innovation and technological development. When companies have the knowledge and technology necessary to innovate processes and products, these challenges are overcome. In this way, we verify that companies are able, besides recycling their products, to modify their design to guarantee the quality of the product in its post-life. This reorganization of the supply chain allows an evolution from closed-loop supply chain to cradle-to-cradle supply chain. This evolution can be achieved through innovation and technological development based on knowledge, allowing sustainability in all processes, as is necessary in the automotive supply chain.

It was observed that some partners are still blocked by the difficulties in recycling their products. For that reason, they are still unable to re-think the entire product cycle, which is the basis for a return supply chain. However, company 3, after managing to overcome the barrier of recycling all their products with quality assurance, is rethinking its entire product design to ensure a pre-defined second life for its products, to ensure their remanufacturing—being one of the companies in the process of executing the cradle-to-cradle supply chain. *"Now we focus on the new organization of all processes and activities associated with the new closed-loop dynamics. Having to rethink the product and how its future will be was one of the biggest challenges."* (Interviewee of company 3).

As mentioned in the methodology, the data collection was done in two phases. The second phase of our data collection was focused on the cradle-to-cradle supply chain. In this second phase, the answers confirmed the main basic issues as sustainability, challenges, and solutions for implementation of return supply chain as referred to in the first phase. It was possible to observe that companies do not have cradle-to-cradle certification,

nor show any desire to obtain it. This internal certification can bring benefits if the products and their design ensure the health and circularity of products. Despite not desiring the certification, they all agree that the basic categories for certification are the same as the ones they use in their business. The companies are acting on the United Nation's 17 sustainable development goals, so their actions may form the basis for cradle-to-cradle. However, the major challenges for the implementation of return supply chain remains the recycling with quality. The difficulty of fully recycling the products and resources is a barrier to fully executing either the closed-loop supply chain or the cradle-to-cradle supply chain.

> *We just don't have cradle-to-cradle certification, although we do everything that is the basis of it. (…) By 2030, we outline the following goals: consider material health, we try to reuse materials, renewable energy, we manage carbon, we try to do water management. As well as, we have social goals, meaning social justice is also a goal. (…) We are trying until at least 2030 that all our resources are something else. The circular economy is our goal. Eco-design is the solution we are missing for this circular economy and cradle-to-cradle.* (Interviewee of company 8).

> *The only thing we are missing now is to get all our products fully recycled, which is cradle-to-cradle certification. With the focus on the circular economy, we were able to get 80% of our resources recycled, but we are aiming for the totality of the product, as well as, all of our products that we are developing to be all recycled and reused in the future.* (Interviewee of company 4).

## 5. Discussion

Our research focuses on understanding the challenges developing a return supply chain in the automotive industry. The strategic business net gives us insight into how sustainability is being developed and fully applied in automotive supply chain companies.

Our results show the companies are responsive to environmental and social issues. It should be noted that in all interviews, none referred to economic sustainability. Moreover, climate change has impacted the automotive industry. Its influence has been studied over time [82]. Climate challenges are proving to be one of the main challenges that companies have to overcome, individually and collectively [83]. The concern with the environment—mainly related to greenhouse gases—has caused reorganization of the automotive industry, that is, changes in vehicle production and in car components, especially in production activities and resource utilization.

Along with climate change, governmental directives have influenced activity in the automotive industry. These influences lead to the modification of all processes in the supply chain, to make them more sustainable. External pressures, along with deadlines, have shown influences on activities and resources use.

Additionally, our results provide evidence that the integration of sustainability into company culture, influences and shapes companies' actions. Sustainability as company culture means that sustainability is integrated in all processes. Sustainability affects the structure and processes of companies [84]. The companies act having regard to environmental, economic and social issues. These effects are spread through business interaction. In the automotive supply chain, the companies are linked by their business relationships. The companies establish relationships between actors and adapt their resources and activities to achieve sustainability. They know that their actions have an impact on other companies, on society and on the environment. Thus, influenced and guided by governmental guidelines, companies have modified their processes. Hence, sustainability affects business interaction, which means, influence on the relationship between actors and their activities and/or resources used. We verified the focus on environmental sustainability, both for reasons of climate issues and governmental guidelines [85], noting that it is on resources and activities that business interaction is more affected. It was possible to observe a focus on the reorganization of resources used and of the activities developed to

achieve the return supply chain. However, those processes have not yet become fully sustainable due to technical difficulties such as total product recycling.

Our research shows that there are several elements that may inhibit and/or enhance the development of return supply chain policies in the automotive industry. In other words, there are elements that facilitate and others that inhibit the interaction between companies to achieve the return supply chain. Our findings show that the enhancers for return supply chains are related with innovation strategies, strategic alliances and governmental policies. On the other hand, we found that the inhibitors to return supply chains are related with organizational and cultural elements. Figure 3 represents both dimensions schematically.

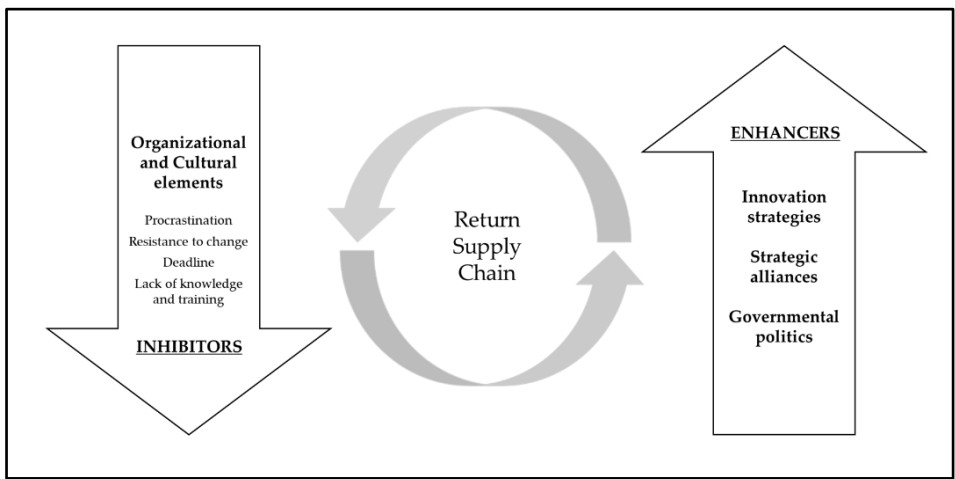

**Figure 3.** Inhibitors and enhancers for developing return supply chain policies.

## 5.1. Inhibitors for Return Supply Chain Development

The inhibitors for return supply chain development are organizational and cultural elements. These elements included procrastination, i.e., the act of delaying or putting off tasks until the last minute. In other words, there is resistance to change. Procrastination can be exacerbated by urgent deadlines to achieve sustainability. Several interviewees spoke of the general resistance (including their own) to change. Even with all the benefits, the external influences, and defined deadlines to work differently, there is resistance. There is a need for new efforts despite the "comfort of the known", which some interviewees refer to as organizational and cultural elements. Hansen et al. (2020) have investigated the resistance to change in companies, finding that resistance can happen at different levels: individual, between companies, in the supply chain, and institutional, leading to different barriers [63]. Our research supports Hansen et al. (2020) and shows that the "country culture" may be a barrier.

Another organizational and cultural element referred to was lack of knowledge and training. However, companies refer to their internal problems with learning capabilities and knowledge as lack of support from their country and the European Union. The lack of knowledge and training make it difficult to accomplish governmental guidelines. Without knowledge, it is not possible to comply with the order, nor to make all processes more sustainable, which is necessary for the total reorganization of the supply chain. According to Kavalić et al. (2021), knowledge promotes the development of the company, economically and financially, as well as being important to achieve the competitiveness and sustainability of a company [86]. This misperception intensifies the resistance to change. The European Union, with the acceptance of the different member states, prepares a set of guidelines with what they should do, but the interviewees say that "it is not enough". In other words, the goals are explained but without specificity on how to achieve them. However, it should be noted that all the stages of implementation of the European Union can

be financed by it, through applications for funds with the needs of the companies in mind. Furthermore, an annual European Unionevaluation is made of what has been achieved in terms of global warming. Each member-state has the autonomy to verify its performance, besides knowing the periodic audits exposed in the document. Zhang et al. (2019) mentioned that the barriers for the execution of the closed-loop supply chain were the lack of government support and laws [87]. However, in this research, despite the interviewees' misperception about guidelines, there is support and monetary access to facilitate the implementation of return supply chains. This support allows firms to invest in technologies which help to improve their knowledge. With it, they can increase the skills and the capabilities necessary to reorganize their activities and resources, to overcome some of the constraints on return supply chain implementation.

The organizational inhibitors elements are easier to overcome. These aspects can be improved by commitment and proactivity of companies.

### 5.2. Enhancers for Return Supply Chain Development

On the other hand, we found several enhancers to develop the return supply chain such as innovation strategy, strategic alliances, and governmental policies that force companies to become more sustainable in several processes.

The business interaction between supply chain partners is essential for the development of sustainable innovative products in the automotive industry [11,88]. There are several studies that present innovation strategies as a solution to achieve sustainability in the supply chain. However, it is necessary to distinguish two aspects of innovation to achieve sustainability. First, for development of the return supply chain, innovation is needed. Second, for total product recycling, innovation is also necessary. The latter constraint can be overcome by technological development of innovative resources and activities leading to rethinking of their entire design, i.e., integration of cradle-to-cradle design framework into the supply chain. There are several studies proposing innovation as an answer to the need for total recycling, but research giving concrete answers as to how process and resources can solve these problems is scarce [89]. The innovation strategies (resource innovation and technological development) are intertwined and are encouraged by the European Union [41]. Tsakalidis (2020) refers to the need for innovative strategies to reduce carbon emissions, either globally (vehicle) or in components (transition of energy systems) [41]. Technological innovation is based on resources and how they can be used and how they are developed [90]. The pressure for innovation has shifted from manufacturers to their suppliers. However, in the case of suppliers with lower financial turnover, the research investment is limited, which makes resource innovation and technological development difficult [88]. These innovative strategies are one of the answers to achieve the goals of the European Union.

The other enhancer for return supply chain development are the strategic alliances. Companies establish partnerships with other companies and/or universities to have access to research and development. Collaborative research and the development of initiatives between universities and industry have become increasingly important in developing process and product innovation [91]. Even in the case of small companies without the capacity to have a research and/or a development center, partnerships with research and development companies and/or universities are a good solutions to achieve sustainability. One of the companies interviewed opened a national contest with the objective of receiving students who would help modify one of the processes in a more sustainable way. This strategic partnership with universities helps to achieve the greater good for both parties. This partnership supports the development of common growth, creating value and increasing the positioning of both the companies and the universities—institution, center, or course in specific areas. The transfer of knowledge from universities to companies contributes to the achievement of goals, being an incentive for a better society with access to more sustainable products. This type of partnership allows companies to expand their network, develop new business relationships, and improve their knowledge. This last

benefit represents a growth of skills and capabilities, referred to previously as barriers for return supply chain implementation.

Finally, government policies have a regulatory role, and at the same time are an enhancer for the implementation of the return supply chain policies in the automotive industry. As previously mentioned, governmental guidelines demonstrate the goals to be achieved, establish timeframes and may provide support and monetary access to the achievement of sustainability goals.

As noted above, if sustainability is embedded in organizational culture, this influences all the processes to needed achieve the return supply chain. In the automotive supply chain, business interaction is difficult as regards product recycling. Some companies still cannot recycle all their products, which means the impossibility of developing a return supply chain. We found that to overcome this barrier, innovation, strategic alliances and governmental politics may be enhancers to return supply chain development. In turn, we found some difficulties, related to organizational and internal elements such as procrastination and lack of knowledge that can, however, be mitigated with the enhancers referred above.

## 6. Conclusions

This paper contributes to the understanding of the sustainability challenges namely, the barriers and solutions to developed return supply chain policies in the automotive industry.

The automotive industry is influenced by climate change and governmental guidelines for implementation of sustainability. Companies create a sustainability-focused organization culture, to make all processes sustainable. It was observed that companies have difficulties in becoming fully sustainable, which increases the challenges for the development of return supply chain policies. We found several inhibiting and enhancer elements for return supply chain development: organizational and cultural elements that reduce return supply chain performance, as opposed to elements such as innovative strategies, strategic alliances, and governmental politicies as enhancers to improve the development of return supply chains.

Our research shows that most of the difficulties hindering the implementation of return supply chains are related to total recycling of the products, as well as to maintaining the quality of the products after recycling. When this challenge was overcome, the actors managed to develop cradle-to-cradle supply chains: they redefined their design and modified the entire supply chain to guarantee an infinite life for the products, and no associated waste. In this situation, the organizational inhibitor elements are easier to be surpassed. These aspects can be improved by commitment and proactivity of companies. Governmental policies can give the economic support for technological innovation, and strategic alliances with universities and/or research centers may give the knowledge and training need to overlap technical barriers.

This study contributed theoretically to increased knowledge about sustainability theory, business interaction and challenges for implementation of return supply chains. Our findings show that when internal organizational and cultural elements and recycling difficulties are overcome, it is possible to develop return supply chain policies in the automotive industry.

Our research also shows managerial contributions, namely the positive role of training and of partnerships between business firms and universities or research centers to support the recycling of the product, and/or to rethinking the supply chain design. Our findings show that the investment in training and/or partnership with universities or research centers may be a way of overcoming the difficulty of recycling the totality of the product and for rethinking the entire supply chain design to achieve return supply chains. The practical contributions for business management are related to establishing new business relationships with service business. With these business relationships, barriers are more easily overcome and automotive services are more capable of being sustainable.

As future research, we propose further investigation into sustainability in any type of automotive supply chain (as was verified in our bibliometric analysis). Another suggestion for further research is an investigation into the internal and external influences for the application of sustainability in companies. Due to the fact that the study was conducted in Portugal, which belongs to the European Union, and covered the automotive sector, it would be of interest to develop other studies to compare and understand the influence of diverse governmental policies in other countries, regions, and supply chains. In addition, more studies are needed on innovation in recycling processes and not only the reference to the need for innovation as a solution to recycling. Finally, we should consider the limitations of this research related to the methodology used, namely the small number of businesses in the strategic business net [92]. Other studies may consider other contexts, case studies and business firms to develop our conclusions.

**Author Contributions:** The research was developed by both authors. Supervision, J.F.P.; Writing—original draft, L.P.S. All authors have read and agreed to the published version of the manuscript.

**Funding:** This paper received funding through research grant UIDB/04521/2020 by FCT—Fundação para a Ciência e Tecnologia (Portugal) for Advance/CSG, ISEG's research center.

**Acknowledgments:** João F. Proença gratefully acknowledges financial support from FCT—Fundação para a Ciência e Tecnologia (Portugal), national funding through research grant UIDB/04521/2020.

**Conflicts of Interest:** The authors declare no conflict of interest.

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
