# Peer review of "Developing Return Supply Chain: A Research on the Automotive Supply Chain"

_sustainability, doi:10.3390/su14116587_

Round 1
Reviewer 1 Report
Congratulations to the Authors. The article is interesting and deals with the important problem of Return Supply Chain in the automotive industry. The authors identified contemporary barriers and challenges of the Return Supply Chain in this industry. I have a few comments:
- The authors presented the ideas of other researchers in the field of Return Supply Chain in an interesting way, however, they did not indicate any specific differences between these approaches, especially as a result of interviews and identified barriers and challenges
- The authors refer to the philosophy of sustainable development, but in the entire content they only emphasize the environmental dimension, why?
- in methodology, the authors should supplement the case study.
- in the discussion of the results, I missed a reference to the results of research on Return Supply Chain (including barriers and challenges) in other countries - which of the identified barriers / challenges are universal and which result from the specificity of the country
Author Response
Author's Reply to the Review Report (Reviewer 1)
Congratulations to the Authors. The article is interesting and deals with the important problem of Return Supply Chain in the automotive industry. The authors identified contemporary barriers and challenges of the Return Supply Chain in this industry. I have a few comments:
Thank you for your review and your comments. I have submitted a new version with the changes made. Following this I will respond to each of your comments.
- The authors presented the ideas of other researchers in the field of Return Supply Chain in an interesting way, however, they did not indicate any specific differences between these approaches, especially as a result of interviews and identified barriers and challenges
Thank you for your comment. Considering the comment, a clarification has been made in the article. Please see lines 289-300.
- The authors refer to the philosophy of sustainable development, but in the entire content they only emphasize the environmental dimension, why?
Thank you for your comment. Considering the data collected and our sample we verify a concern with the economic and social development, however, the companies show no difficulty in these dimensions. It was in the environmental dimension that companies demonstrated their greatest concern, and we focused on it. The limitations of the environmental dimension caused challenges to the implementation of the return supply chain, which did not occur in the social and economic dimension. On the other hand, as referred in lines 177-180: “Environmental sustainability is the sustainability dimension which European Union give focus more attention. For that reason, it has undergone the most changes, since the changes in resources are able to reduce the impact of their activities on the environment.”
- in methodology, the authors should supplement the case study.
Thank you for your comment. Considering the comment, a clarification has been made in the article.
- in the discussion of the results, I missed a reference to the results of research on Return Supply Chain (including barriers and challenges) in other countries - which of the identified barriers / challenges are universal and which result from the specificity of the country
Thank you for your comment. Considering the comment, a clarification has been made in the article. Please see lines 463 – 469.

Reviewer 2 Report
The paper is well written and interesting to read.
Author Response
Author's Reply to the Review Report (Reviewer 2)
The paper is well written and interesting to read.
Thank you for reviewing the article and for your nice words.
Reviewer 3 Report
This paper examines the challenges, obstacles, and solutions faced by the automotive industry in developing a return supply chain through a literature review, econometric analysis, and company interviews. The entire paper is logically structured and informative. For example, the paper begins with showing the need for the automotive industry to develop cradle-to-cradle supply chains to restructure their supply chains by discussing the positive impact of people on climate change during the epidemic, and the ELV directive. And, the need for this study is demonstrated by showing the research gap in the development of cradle-to-cradle supply chains in the automotive industry through the method of econometric analysis.
The following are some of the queries and suggestions from the reading process.
- The study surveyed 12 companies to extract key data from the content of the interviewees. However, the total number of respondents and the number of respondents in different capacities are not clearly mentioned in the text. In other words, what was the sample size? Was one person interviewed within each company? Are all interviewees' data valid? How was the data pre-processed?
- Line 594 is mislabeled
- The Discussion section could be further separated by subsections to make the structure of the discussion clearer. For example, 5.1 Inhibitors 5.2 Enhancement…
- Are the interviewees the same for both interviews?
- The design idea of the interview script needs to be clarified. What is the connection among different questions?
- The main contribution of this paper is to help automotive companies find challenges in developing cradle-to-cradle supply chains, but the contribution to solutions is limited. Although the theoretical contribution is large, the practical contribution to business management is insufficient.
Author Response
Author's Reply to the Review Report (Reviewer 3)
This paper examines the challenges, obstacles, and solutions faced by the automotive industry in developing a return supply chain through a literature review, econometric analysis, and company interviews. The entire paper is logically structured and informative. For example, the paper begins with showing the need for the automotive industry to develop cradle-to-cradle supply chains to restructure their supply chains by discussing the positive impact of people on climate change during the epidemic, and the ELV directive. And, the need for this study is demonstrated by showing the research gap in the development of cradle-to-cradle supply chains in the automotive industry through the method of econometric analysis.
Thank you for your review and your comments. We have submitted a new paper version according to your notes. Following this we will respond to each of your comments.
The following are some of the queries and suggestions from the reading process.
- The study surveyed 12 companies to extract key data from the content of the interviewees. However, the total number of respondents and the number of respondents in different capacities are not clearly mentioned in the text. In other words, what was the sample size? Was one person interviewed within each company? Are all interviewees' data valid? How was the data pre-processed?
Thank you for your comment. Considering this comment, a clarification has been made in the article.
- Line 594 is mislabeled
Thanks for your comment. It was made the correction.
- The Discussion section could be further separated by subsections to make the structure of the discussion clearer. For example, 5.1 Inhibitors 5.2 Enhancement…
Thanks for your comment. Your comment has been taken in account, and we create different subsections.
- Are the interviewees the same for both interviews?
Thank you for your question. Considering your doubt, a clarification has been made in the article. The interviewees were the same, having been interviewed in two different phases. The second phase focused on delving deeper into the barriers and solutions for the implementation of the return supply chain.
- The design idea of the interview script needs to be clarified. What is the connection among different questions?
Thank you for your comment. Considering your comment, a clarification has been made in the article. The interviews were semi-structured and semi-directional considering the script. The different questions of the script intended to understand the challenges, namely, the barriers and solutions to develop return supply chain policies in the automotive industry. Thus, we started with sustainability for the company and the reasons for its implementation. After that, we explored how sustainability was verified in the three dimensions internally, in the company and externally, with the companies in the supply chain. Following, we verify the solutions and barriers to be fully sustainable, that is, in order to implement the return supply chain.
- The main contribution of this paper is to help automotive companies find challenges in developing cradle-to-cradle supply chains, but the contribution to solutions is limited. Although the theoretical contribution is large, the practical contribution to business management is insufficient.
Thank you for your comment. Considering the note, the managerial contribution has been included in the paper. Please see lines 895-900.

Round 2
Reviewer 3 Report
Thank you very much for the revision. I think this article has met the criteria for acceptance. The theoretical basis of this paper is rich, and the empirical process is logical, and the research results can contribute to the development of return supply chain policies of enterprises. Except for some simple adjustments to the sentence patterns, there are no major problems in other parts.